# Combining Celiac and Hepatic Vagus Nerve Neuromodulation Reverses Glucose Intolerance and Improves Glycemic Control in Pre- and Overt-Type 2 Diabetes Mellitus

**DOI:** 10.3390/biomedicines11092452

**Published:** 2023-09-04

**Authors:** Jonathan J. Waataja, Anders J. Asp, Charles J. Billington

**Affiliations:** 1ReShape Lifesciences Inc., Irvine, CA 92618, USA; 2Department of Physical Medicine and Rehabilitation, Mayo Clinic, Rochester, MN 55605, USA; 3Minnesota Veterans’ Administration Medical Center, Minneapolis, MN 55417, USA

**Keywords:** type 2 diabetes, vagus nerve, vagus nerve stimulation, bioelectronics, neuromodulation, neurological disorders, neuropathy, glycemic dysregulation, heart rate variability and vagal tone

## Abstract

Neurological disorders and type 2 diabetes mellitus (T2DM) are deeply intertwined. For example, autonomic neuropathy contributes to the development of T2DM and continued unmanaged T2DM causes further progression of nerve damage. Increasing glycemic control has been shown to prevent the onset and progression of diabetic autonomic neuropathies. Neuromodulation consisting of combined stimulation of celiac vagal fibers innervating the pancreas with concurrent electrical blockade of neuronal hepatic vagal fibers innervating the liver has been shown to increase glycemic control in animal models of T2DM. The present study demonstrated that the neuromodulation reversed glucose intolerance in alloxan-treated swine in both pre- and overt stages of T2DM. This was demonstrated by improved performance on oral glucose tolerance tests (OGTTs), as assessed by area under the curve (AUC). In prediabetic swine (fasting plasma glucose (FPG) range: 101–119 mg/dL) the median AUC decreased from 31.9 AUs (IQR = 28.6, 35.5) to 15.9 AUs (IQR = 15.1, 18.3) *p* = 0.004. In diabetic swine (FPG range: 133–207 mg/dL) the median AUC decreased from 54.2 AUs (IQR = 41.5, 56.6) to 16.0 AUs (IQR = 15.4, 21.5) *p* = 0.003. This neuromodulation technique may offer a new treatment for T2DM and reverse glycemic dysregulation at multiple states of T2DM involved in diabetic neuropathy including at its development and during progression.

## 1. Introduction

There is a bi-directional relationship between the development and progression of autonomic dysfunction and glycemic dysregulation involved with type 2 diabetes mellitus (T2DM). Detrimental changes in autonomic tone precede development of T2DM [1,2,3,4] and progression of T2DM leads to further negative autonomic states [5,6,7,8]. The activity of the vagus nerve plays a central role in this process. Cardiac function and heart rate variability (HRV) have been widely used as surrogate measurements of vagus nerve activity, autonomic tone and autonomic dysfunction [9,10,11]. Changes in HRV precede development of T2DM, and T2DM progression further leads to changes in HRV in a feed-forward fashion. 

Conditions that lead to the development of T2DM are associated with decreased HRV in non- and pre-diabetics. This is demonstrated through the following correlations: decreased HRV with higher levels of insulin resistance in non-diabetics [12,13,14,15], decreased HRV in non-diabetics with decreased β-cell function [5,16], lower HRV in non-diabetics with hyperinsulinemia [10], and low HRV in pre-diabetics [9].

Glycemic dysregulation following the development of T2DM leads to decreased HRV [17]. For example, in type 2 diabetics, increased HbA1c, fasting plasma glucose (FPG), and 2 h post-load plasma glucose levels during OGTTs correlate with decreased HRV [9]. Decreased HRV is also associated with T2DM progression. A clinical study by John et al. [18] categorized the T2DM disease state as duration from disease diagnosis. They divided the subjects into 3 groups having different diabetic durations: <5 years, 5–10 years and >10 years. Disease duration greater than 10 years demonstrated the greatest changes in HRV [18].

In addition to autonomic neuropathy, unregulated T2DM has been correlated with many other neurological disorders, including peripheral non-autonomic neuropathy (e.g., somatosensory neuropathy), retinopathy, and Alzheimer’s disease [19,20,21]. Cardiovascular disease is also associated with diabetes-induced autonomic dysfunction [17,22]. Methods to treat glycemic dysregulation in the early stages of T2DM have been shown to help reverse diabetic neuropathy [19], including autonomic neuropathy [23]. In the early stages of T2DM, lifestyle modification (such as healthy diet and exercise) is a strategy to prevent neuropathy [19]. However, 75% of type 2 diabetics do not adhere to lifestyle modification [24]. Many clinical trials have demonstrated that GLP-1 receptor agonists (GLP-1 RAs) are highly effective at restoring glycemic homeostasis and weight loss. GLP-1 RA-induced HbA1c reduction ranges from 0.8 to 1.9% [25], and weight loss up to 12% from baseline [26]. This class of drugs has been used to treat T2DM in the early and advanced stages [27]. However, as observed with many T2DM medications, adherence is a major problem with GLP-1 RAs with reported discontinuation in real-world scenarios to be approximately 45% at 1 year [28,29] due to unwanted side effects [30], high cost [31], and unwillingness to follow the physician’s treatment advice [32]. Poor adherence has been demonstrated in numerous clinical studies (Table 1).

Many non-pharmaceutical techniques are under development, including next generation insulin pumps, gene therapy, duodenal modification including mucosal resurfacing and stimulation, peripheral-focused ultrasound (pFUS), and various vagus nerve modulations. Research has shown that these have promise as possible treatments for T2DM, but they have limitations. Closed-loop insulin pump technology has not been well adopted by type 2 diabetics for many reasons, one of which is the large amount of insulin required for therapy for type 2 diabetics compared to type 1 diabetics (T1DM): approximately 100 Units per day compared to 10 [37]. This requires frequent refilling of the reservoir. Another significant barrier is the high out-of-pocket cost: Medicare does not currently reimburse closed-loop technology for treating T2DM. Consequently, less than 0.1% of type 2 diabetics use insulin pump technology [38]. 

Genetic modulation has shown promise in preclinical experiments [39,40,41] to increase glycemic control. However, current gene therapy has a clinical risk, and the accepted types of diseases under investigation are severe, rare, and debilitating [42]; not metabolic disorders. Duodenal mucosal resurfacing (DMR) has shown positive results in early clinical trials [43], but the endoscopic DMR procedure carries potential risks of perforation of the intestine and intestinal leakage [44]. Duodenal stimulation of the subserosal layer of the anterior duodenal wall has demonstrated improved glycemic control in preclinical [45] and clinical studies. In an open-labeled single-arm study in 12 obese T2DM subjects, HbA1c decreased by 0.8% (from a baseline of 8%) and FPG decreased by 33 mg/dL (from a baseline of 173 mg/dL) after 12 months of stimulation [46]. Further blinded and adequately powered studies are needed to determine the effectiveness of this approach. 

Glycemic control may be improved through pFUS of the hepatoportal nerve plexus as demonstrated by experiments in rodent models of T2DM and in swine [47]. However, considerable technological advancements are required for at-home application because physicians and technicians are currently required to operate the system. Flexible body-conformal hand-held ultrasound devices are under development [48], which may help bridge this technical gap. However, their focal depth of 10–20 mm does not reach the hepatoportal nerve plexus in humans. 

External and internal vagus nerve stimulation has demonstrated effects on glycemia. A significant focus of external vagus nerve stimulation has been on transcutaneous auricular vagus nerve stimulation (taVNS). The stimulation site is on the external ear, near the auricular branch of the vagus nerve (ABVN). Studies demonstrate taVNS may be an effective treatment for epilepsy and depression [49,50], and it has shown positive results as a treatment for T2DM in rodent models for T2DM [51]; however, this has not translated well in clinical settings, where there has been a modest increase in glycemic control or none at all [52,53,54]. Internal stimulation of the vagus nerve and its branches has shown more promise in regulating glycemia.

In a study by Meyers et al. [55], increased glycemic control was observed with stimulation of the distal end of the ligated left cervical vagus in non-diabetic anesthetized rats. Despite positive results, this method would not be clinically applicable due to ligation of the cervical vagus nerve. A similar study found the same results by assessing rat cervical stimulation using intraperitoneal glucose tolerance tests (IPGTTs) [56]. Selective efferent vagal nerve stimulation of the anterior sub-diaphragmatic vagal trunk in a rat model of T2DM demonstrated positive results in increasing glycemic control [57]. This study applied a high-frequency blocking current cranially to a stimulation electrode for selective efferent stimulation. This has clinical practicality compared to the Meyers et al. approach in that it did not require nerve ligation. The study reported improvements during OGTTs but did not report a decrease in FPG. In a study by Yin et al. [58], glycemic control was enhanced by dorsal sub-diaphragmatic vagus stimulation in a T2DM rat model. Glucose tolerance tests demonstrated increased glycemic control, but no change in FPG was reported. Radiofrequency ablation of the vagus hepatic branch has shown increased glycemic control in glucose-intolerant rats [59]. However, hepatic vagotomy causes increased hypoglycemic episodes, adverse changes in feeding behavior, negative effects on liver regeneration, and increased metastasis during liver cancer [60,61,62,63]. 

Since vagal nerve branches control multiple organ systems involved in plasma glucose (PG) regulation, multi-branch vagal modulation may be necessary to maximize the effects on glycemic regulation. We previously demonstrated that a novel technique of electrically blocking conduction through the hepatic branch of the vagus nerve innervating the liver with simultaneous stimulation of the celiac branch innervating the pancreas improved glycemic control in animal models of T2DM [64]. Electrical block involved the application of a high-frequency alternating current (HFAC @ 5000 Hz), which has been shown to reversibly block conduction through the sub-diaphragmatic vagus nerve [65]. Applying HFAC to the hepatic branch of the vagus nerve with concurrent stimulation (bi-phasic at 1 Hz frequency) of the celiac branch will be referred to as HFAC + stimulation. Experiments in the Zucker rat model of T2DM demonstrated that HFAC + stimulation increased glycemic control and mimicked hepatic vagotomy + stimulation. The dual procedure was superior to standalone stimulation or vagus blockade [64]. In chronic experiments on alloxan-treated T2DM swine, HFAC+stimulation demonstrated increased glycemic control according to oral glucose tolerance tests (OGTTs) and FPG measurements. The swine were glucose intolerant but not insulin-dependent, attributes of a T2DM state. 

The alloxan-treated swine in the previous HFAC + stimulation experiments would be characterized as having attributes of a high pre-diabetic/low diabetic state (FPG = 120 ± 14 mg/dL). However, effective treatments at more distinct stages of glucose intolerance may be desired to demonstrate the potential of HFAC + stimulation to treat T2DM at the early and later stages of the disease. In this study, we tested if HFAC + stimulation can increase glycemic control in alloxan-treated swine that had attributes of pre-diabetes and diabetes. OGTTs and observed changes in FPG were used to test for changes in glycemic control during and following delivery of HFAC + stimulation. 

## 2. Materials and Methods

The experiments were approved by the Institutional Animal Care and Use Committees at the University of Minnesota (Minneapolis, MN, USA) and North American Science Associates, Inc. (Brooklyn Park, MN, USA)

### 2.1. High Frequency Alternating Current (HFAC)

A Viking Model 2002 neuroregulator (ReShape Lifesciences Inc., Irvine, CA, USA) was used to generate a 5000 Hz HFAC signal. The signal consisted of a constant current bi-phasic square waveform consisting of charge and recharge components (Figure 1a). The charge and recharge waveforms were generated from the same current source so the current in each component matched. Shorting periods (10 µs), in which the electrodes were short-circuited together, were incorporated as part of the duty cycle of each of the charge and recharge waveforms to remove any charge remaining after application of the waveform. The pulse generator was assessed for direct current (DC) and met a <1 µA leakage specification. 

### 2.2. HFAC + Stimulation Experiments

In adult Yucatan swine (~45 kg, n = 6), a proprietary titrated dose of alloxan was administered at Sinclair Bio Resources (Auxvasse, MO, USA). The swine, which were not insulin dependent, were monitored and fed ad libitum for 24 h following treatment to prevent any possible hypoglycemia due to the release of insulin caused by β-cell death. The swine were monitored for 8 weeks prior to shipment (Figure 2). Swine were offered food twice per day (Teklad 7200, Envigo, for the non-diabetic swine and CU Sinclair S-9 Ration, Sinclair Bio Resources, for the alloxan-treated swine) except for an 18 h fast prior to glucose challenges. An injection of 600 mg/kg glucose was administered at the initiation of intravenous glucose tolerance tests (IVGTTs), and blood was sampled with a glucometer (One Touch Ultra, LifeScan, Malvern, PA) at baseline and 3, 5, 7, 10, 15, 30, 45, 60 90 and 120 min following injection. 

Swine were trained to wear a jacket to house mobile charging units for future charging sessions and to drink 100 mL of diet Gatorade delivered through a syringe. Oral consumption of 75 g of glucose dissolved in 100 mL of diet Gatorade was administered at the start of the OGTTs, which are widely used to assess glycemic control in swine [66,67].

For surgical implantation of the device, the swine were anesthetized with Telazol/Xylazine given IM at a dose of 6 mg/kg and 1 mg/kg Xylazine and intubated and maintained on isoflurane (1.0–2.0% to effect). Two Viking neuroregulators (ReShape Lifesciences Inc., Irvine, CA, USA) and four Viking Model 2200-47E leads (ReShape Lifesciences Inc., Irvine, CA, USA) were implanted. Two Viking leads with platinum-iridium cuff electrodes, which made a 180° contact with the nerve, were placed on the posterior sub-diaphragmatic vagal trunk cranial nerve to the celiac branching point (referred to as the celiac branch) and sutured to the esophagus to deliver a bi-phasic charge-balanced pulse at 1 Hz (Figure 1b, 4 ms pulse width and 8 mA current amplitude). A second pair of identical cuff electrodes were placed on the anterior sub-diaphragmatic vagal trunk cranial nerve at the hepatic branching point (referred to as the hepatic branch) and sutured to the esophagus to deliver HFAC (5000 Hz, 8 mA current amplitude). During the experiments, 5000 and 1 Hz were applied concurrently (waveforms illustrated in Figure 2). The location of electrodes on the vagus nerve are illustrated in Figure 3. The leads were tunneled to two Viking neuroregulators in a subcutaneous pocket above the ribcage on either side of the swine. Each lead had a wing suture tab between the lead tip and the neuroregulator, which was sutured to the stomach as a strain relief. 

Ten days following implantation, OGTTs were performed. Swine with FPGs > 126 mg/dL were considered diabetic [68]. The others demonstrated mild glucose intolerance (FPG ≥ 100 mg/dL and <126) mg/dL and are referred to as pre-diabetic. Sham conditions consisted of the devices implanted but not delivering HFAC + stimulation signals.

### 2.3. Analysis

Baseline glucose was measured at 10 ± 3 min prior to OGTTs and IVGTTs. The glucose response to tolerance tests was quantified by calculating the area under the curve (AUC, (glucose concentration (mg/dL) × time)/1,000 = area units (AUs)) [69,70,71]. Details of calculating the AUC have been described previously [64], but briefly, the line connecting two subsequent data points and the x-axis was calculated as one segment. The total number of segments following the glucose challenge was then summated. The AUC involved a net calculation, meaning that area under the x-axis was subtracted from the area above the x-axis In some cases, percent change of the PG response was utilized to normalize responses to their baseline value. The normalized area under the curve of was calculated in a net manner. Normalized AUC = (PG percent change*time)/1,000 = percent area units (%AUs). Comparisons between the condition tested and the sham consisted of a Mann–Whitney U test where a nominal alpha level of 0.05 was considered significant. There were no adjustments for multiplicity. All data were presented as mean ± SEM, unless indicated otherwise. Boxplot horizontal lines represent data as median; box ends first and third quartile, and whiskers as maximum and minimum.

## 3. Results

### 3.1. Alloxan Treated Swine Demonstrated Pre-Diabetes and Diabetes 

Alloxan was used to create non-insulin dependent glucose intolerant swine for chronic HFAC + stimulation studies. Conditioning animals with alloxan is a well-established and widely used method for inducing diabetes by destroying β-cells [72,73,74,75,76,77,78]. Alloxan enters the β-cells of the islet through uptake by the GLUT2 transporter and establishes a redox cycle by forming superoxide radicals that undergo dismutation to hydrogen peroxide. Thereafter, highly reactive hydroxyl radicals are formed by the Fenton reaction. The action of reactive oxygen species with a simultaneous massive increase in cytosolic calcium concentration causes rapid destruction of the β-cells, which decreases insulin production causing glucose intolerance. 

Following alloxan treatment, fasting plasma glucose (FPG) ranged from 101–207 mg/dL. The swine were divided into two groups: pre-diabetic (FPG ≥ 100 mg/dL and <126 mg/dL) and diabetic (FPG ≥ 126 mg/dL) [68]. Three swine had FPGs between 100 and 124 mg/dL (Range: 101–119 mg/dL; average = 113 ± 4 mg/dL), and 3 with FPG greater than 125 mg/dL (Range: 133–207 mg/dL; average = 167 ± 15 mg/dL). There was a significant decrease in performance during IVGTTs prior to and following alloxan treatment for both groups (Figure 4a–d) as quantified by AUC measurements (Pre-diabetic: the median AUC pre-alloxan = 11.0 AUs (IQR = 10.1, 11.8); the median AUC post-alloxan = 18.7 AUs (IGR=16.5, 21.4); *p* < 0.01 Mann–Whitney U Test. Diabetic: the median AUC pre-alloxan = 11.6 AUs (IQR = 10.6, 12.3); the median AUC post-alloxan = 22.8 AUs (IQR
= 18.9, 33.3); *p* < 0.05 Mann–Whitney U Test). 

Reduced performance on IVGTTs as assessed by AUC may be confounded by changes in baseline following alloxan in both the pre-diabetic and diabetic groups. To address this PG was normalized to baseline during IVGTTs for both pre-diabetic and diabetic groups (Appendix A) which yielded similar results. 

### 3.2. High Frequency Alternating Current + Stimulation Increased Performance on OGTTs and Decreased FPG in Pre-Diabetic and Diabetic Swine

Oral glucose tolerance tests (OGTTs) were conducted to assess glycemic control during HFCA + stimulation in pre-diabetic and diabetic swine. In the pre-diabetic swine, under sham conditions, PG increased to 150 ± 11 mg/dL by 30 min into the OGTT (Figure 5a, baseline = 113 ± 4 mg/dL), which stayed constant through 2 h (2 h PG = 145 ± 6 mg/dL). During sham OGTTs in the diabetic swine, PG increased over time and peaked at 282 ± 60 mg/dL 2 h following glucose ingestion (Figure 5c, baseline = 167 ± 15 mg/dL). Plasma glucose recovered to baseline in both groups at the cessation of the OGTT experiment (i.e., 4 h after the ingestion of glucose). As assessed by the AUC of the OGTTs [69,70,71], there was a significant increase in performance during OGTTs in pre-diabetic and diabetic swine with the application of HFAC + stimulation. 

Following HFAC + stimulation, AUC significantly decreased to a median AUC of 15.9 AUs (IQR = 15.1, 18.3) from a median AUC of 31.9 AUs (IQR = 28.6, 35.5) at baseline (*p* = 0.004, Mann–Whitney U Test) in pre-diabetic swine (Figure 5b) as well to a median AUC of 16.0 AUs (IQR = 15.4, 21.5) from a median AUC of 54.2 AUs (IQR = 41.5, 56.6) at baseline, (*p* = 0.003, Mann–Whitney U Test), in diabetic swine (Figure 5d). The AUC of the OGTTs with HFAC + stimulation was the same for diabetic and pre-diabetic groups.

Decreased baseline during HFAC + stimulation conditions may confound the interpretation of glycemic control as assessed by the above AUC analysis. To address this PG was normalized to baseline during OGTTs (Appendix A) which yielded similar results as absolute PG AUC analysis. 

Finally, in both groups, FPG decreased significantly from either a pre-diabetic or diabetic state to a non-diabetic state two days following application of HFAC + stimulation (Pre-diabetic FPG following HFAC + stimulation = 70 ± 3 mg/dL (range: 60-86 mg/dL), from a baseline FPG = 113 ± 4 mg/dL (range: 101–119 mg/dL), *p* < 0.01, Diabetic FPG following HFAC + stimulation = 70 ± 2 mg/dL (range: 60–82 mg/dL) from a baseline of FPG = 167 ± 15 mg/dL (range: 133–207 mg/dL), *p* < 0.01, Mann–Whitney U Tests). In non-alloxan treated/non-diabetic swine, there was no change in FPG following HFAC + stimulation (baseline FPG = 81 ± 3 mg/dL (range: 73–89 mg/dL), FPG following HFAC + stimulation applications = 79 ± 4 mg/dL (range: 73–88 mg/dL)). There was no indication of hypoglycemia during these experiments. Results indicating increased glycemic control during OGTTs had already been reported for non-alloxan treated swine [64]. 

Sub-diaphragmatic vagus nerve neuromodulation has been shown to induce weight loss and increase glycemic control in clinical studies [79]. The swine in this study did not experience weight loss, and weight change can be excluded as a possible mechanism in this study.

## 4. Discussion

These results demonstrated a dichotomy in fasting plasma glucose (FPG) following alloxan administration in swine. Half of the swine demonstrated characteristics of mild glucose intolerance/pre-diabetes (FPG between 101–119 mg/dL, classification defined in methods) and half demonstrated characteristic of diabetes (as defined by FPG ≥ 126 mg/dL [68]). Increased glycemic control was evident through the improved performance on OGTTs during HFAC + stimulation in the pre-diabetic and diabetic groups. FPG was also significantly reduced to non-diabetic levels following HFAC + stimulation in both groups.

Results from this study have implications for the treatment of diabetic autonomic dysfunction by using HFAC + stimulation. Autonomic dysfunction develops during early glycemic dysregulation, including during pre-diabetes [9,80]. Here we demonstrated that HFAC + stimulation can reverse a pre-diabetic state to a non-diabetic state. 

Increasing glycemic control in the early stages of overt diabetes has been shown to reverse diabetic automimic neuropathy [23]. Autonomic dysregulation increases with the progression of T2DM [9]. This study also demonstrated reversal of glycemic dysregulation from an overt diabetic state to a non-diabetic state following HFAC + stimulation. Our previous study [64] demonstrated that HFAC + stimulation improved glycemic dysregulation in non-diabetic swine. This suggests that the therapy window for HFAC + stimulation may start prior to the development of pre-diabetes (Figure 6) where there are signs of mild insulin resistance and a decrease of vagal tone.

Not only is low HRV a surrogate for measuring cardiac autonomic tone, but it also correlates with sub-diaphragmatic vagus nerve tone. For example low HRV is associated with T2DM-induced gastroparesis [81], indicating low vagal gastric branch tone. Experimental evidence also suggests that low HRV is a surrogate for indicating low vagal tone of the vagus branches that innervate organs involved with PG regulation. This was demonstrated by the correlation of low HRV with insulin resistance [12,13,14,15] and with decreased β-cell function [5,16]. Also, insulin resistance and decreased β-cell function are correlated with autonomic dysregulation outside of hyperglycemic mechanisms [82,83]. Type 2 diabetic treatments that improved glycemic outcomes also reversed low HRV [84]. 

The results from this study demonstrated glycemic improvements beyond other investigations involving vagus nerve modulation as a method of increasing glycemic control. Our findings showed that the dual approach of HFAC + stimulation caused considerable (167 ± 15 mg/dL to 70 ± 3 mg/dL), and prolonged improvements in FPG. This had not been seen in previous studies, at least to this extent, that utilized stand-alone sub-diaphragmatic vagus stimulation [58] or stand-alone hepatic ligation [85]. Enhanced performance on OGTTs with HFAC + stimulation suggests that HFAC + stimulation may additionally improve glycemic control in the postprandial period. 

There were limitations to this study. First, the number of animals studied was relatively low (n = 6). Future research will include highly powered longitudinal studies to test for safety and efficacy durability. Also, this study will serve as a basis for ensuring appropriate statistical power for future studies. Second, this study did not explore the mechanisms behind the effects of HFAC + stimulation. Future studies will measure hormones such as insulin, glucagon, c-peptide and GLP-1 during IVGTTs and OGTTs. Finally, HRV and HRR were not measured. These cardiac outcomes will be assessed in chronic stimulation studies. 

It is important to note that this study does not propose the development of T2DM to be simply the product of autonomic dysregulation but occurs in parallel with other factors such as excess weight, lifestyle (including poor diet and little exercise) and genetics [86,87,88]. It does propose that the product of these factors on top of autonomic dysfunction leads to glycemic dysregulation, contributing to further autonomic dysfunction and T2DM progression in a feed-forward loop (Figure 7). Treatments to control glycemic dysregulation in many stages of T2DM that lead to an exit of the loop and a sustained euglycemia state (Figure 7) are needed. The results from our studies support the potential of using HFAC + stimulation to cause increased glycemic control at different levels of glycemic dysregulation.

Furthermore, this paper does not suggest that HFAC + stimulation signals directly affect autonomic dysregulation but that may have indirect effects through improved glycemic control. Treatments that reverse glycemic dysregulation at different severities of the disease correlate with autonomic function restoration [84,89]. HRV and systemic autonomic function may eventually be indirectly improved through more time in a euglycemic state. However, future studies are needed for a longitudinal characterization of the recovery of systemic autonomic function with the HFAC + stimulation methodology. 

Our previous research [64] found that glycemic control was enhanced during OGTTs in non-alloxan-treated swine with an FPG = 68 ± 3 mg/dL. This demonstrated that HFAC + stimulation was effective in three categories of glycemic control: non-diabetic, pre-diabetic and diabetic. There was no change in FPG following the application of HFAC + stimulation in the non-alloxan treated swine, suggesting that HFAC + stimulation was only effective in lowering FPG in glycemic-compromised animals. Behavior associated with hypoglycemia in swine (fatigue, tics/tremors, ataxia and seizures) [90] was not observed following HFAC + stimulation in non-alloxan-treated swine. 

Variability in the time between alloxan treatment and the post-alloxan IVGTT may explain the increase in FPG during IVGTT experiments in this study compared to our previous study [64]. However, during sham OGTT experiments in the previous study, FPG was 120 ± 14 mg/dL which is similar to the sham FPGs during the sham OGTTs observed in this study (pre-diabetic: 113 ± 4 mg/dL, diabetic: 136 ± 12 mg/dL).

Safety was not assessed in this study, however, applications of the same HFAC and stimulation parameters used in this study revealed a healthy vagus nerve at the site of the cuff electrodes following 9 applications of HFAC + stimulation in our previous study [64]. There was no adverse behavior in the swine during these studies or indications of hypoglycemia. 

A closed-loop system to blunt glucose spikes on demand may be realized in a system using HFAC + stimulation neuromodulation. Since glucose was offered 5 min prior to the start of OGTTs, HFAC + stimulation the system may work with continuous glucose monitoring to initiate delivery of signals during glucose spikes. Finally, the system may be integrated with machine learning and AI to optimize HFAC + stimulation parameters over time.

## 5. Conclusions

Up to 42% of type 2 diabetics experience neuropathy [91], which may increase endocrine dysfunction and lead to disease progression. Increasing glycemic control is the most effective method for delaying or reversing diabetic neuropathy. Despite new T2DM therapeutic options, such as GLP-1 RAs, there is still a need for therapies tailored to a patient’s compliance that are cost effective over the long-term and decrease unwanted side effects. Targeted electrical vagal neuromodulation at sites close to organs that regulate PG may address these issues. HFAC + stimulation presents a uniquely adjustable therapeutic experience by offering a personalized T2DM treatment method that has the capacity to change with the vast number of parameters involved with multisite/multi-frequency vagal neuromodulation. Future research includes safety and efficacy studies on swine (on the order of several months) that are suitable for gaining regulatory approval for a human trial.

## 6. Patents

US Patents: 8483830, 9333340 and additional related foreign patents.

Additional patents pending.

## Figures and Tables

**Figure 1 biomedicines-11-02452-f001:**
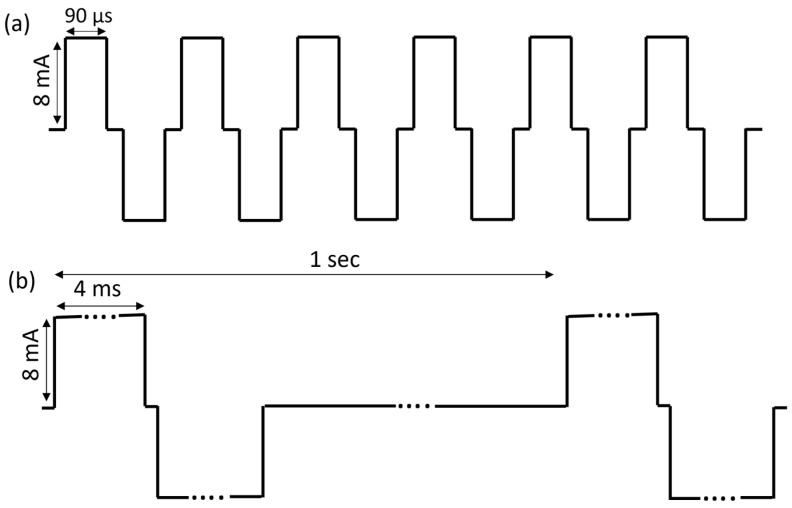
Two Viking neuroregulators were used to generate high and low frequency signals. The high frequency consisted of a 5000 Hz HFAC charge-balanced waveform (**a**). The low-frequency signal consisted of a 1 Hz charge-balanced waveform (**b**).

**Figure 2 biomedicines-11-02452-f002:**
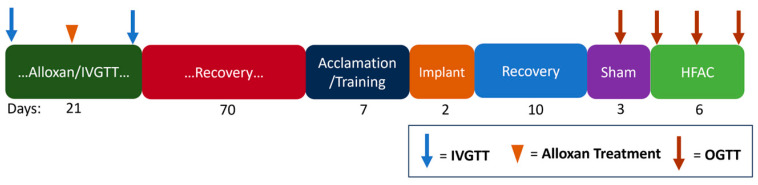
Experimental timeline highlights the long duration of time between alloxan treatment and the initiation of experiments. This gave ample time for a static glycemic state in swine following alloxan treatment prior to OGTT experiments. Blue arrows indicate IVGTTs, orange arrowhead indicates alloxan treatment and orange arrows indicates OGTTs.

**Figure 3 biomedicines-11-02452-f003:**
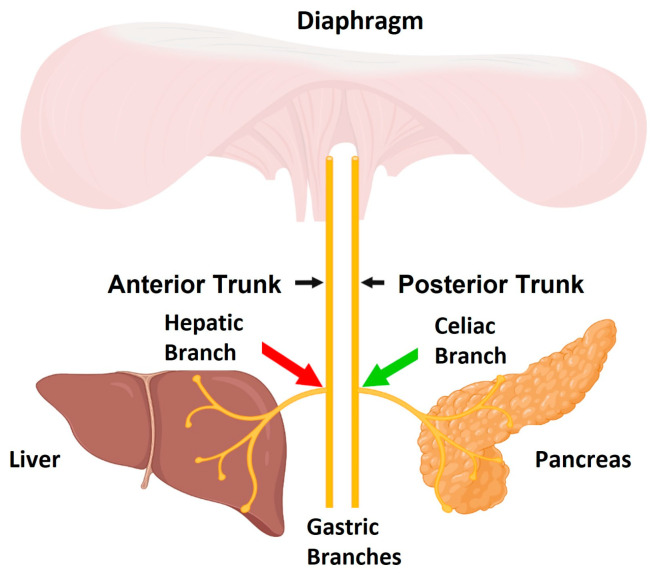
HFAC was delivered to the hepatic branches (red arrow) and stimulation was applied to the celiac branch (green). Modulation of both branches has demonstrated modified liver and pancreatic function leading to changes in PG. Figure adapted from Waataja et al. 2022 [64] and originally created using biorender.com (accessed on 29 June 2022).

**Figure 4 biomedicines-11-02452-f004:**
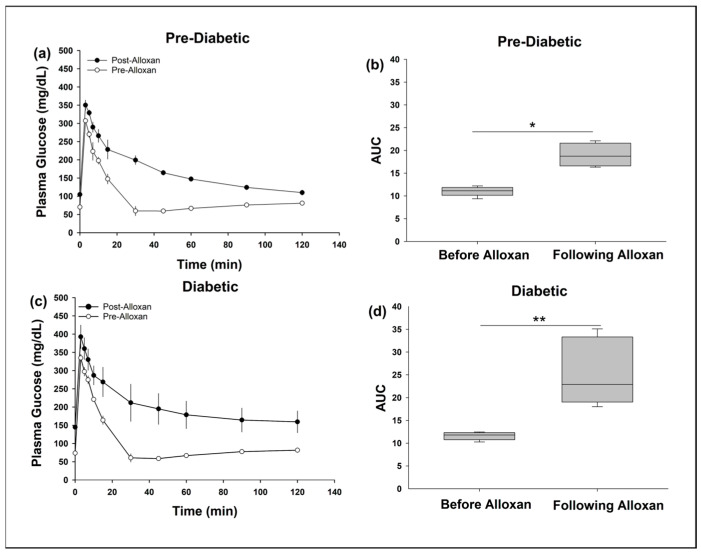
Intravenous glucose tolerance tests demonstrated glucose intolerance in Pre-Diabetic (**a**,**b**) and Diabetic (**c**,**d**) swine. This was demonstrated through higher FPG and greater area under the curve (AUC) following alloxan in both groups. * *p* < 0.01, ** *p* = 0.012.

**Figure 5 biomedicines-11-02452-f005:**
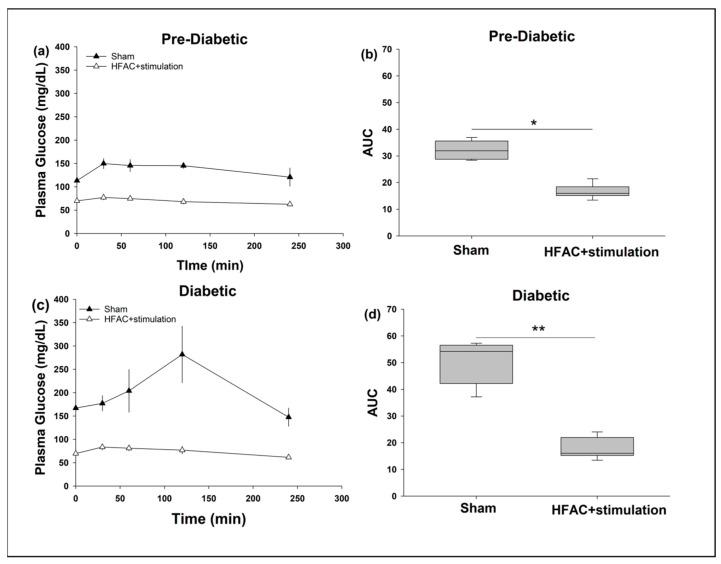
Application of HFAC + stimulation increased performance on OGTTs in Pre-Diabetic (**a**,**b**) and Diabetic (**c**,**d**) swine. HFAC+ stimulation were applied concurrently starting at 5 min following glucose ingestion and for the entire duration (4 h) of the OGTT experiment. Note: A baseline decrease during HFAC + stimulation experiments was due to lasting attenuation of FPG with HFAC + stimulation applications * *p* = 0.004, ** *p* = 0.003.

**Figure 6 biomedicines-11-02452-f006:**
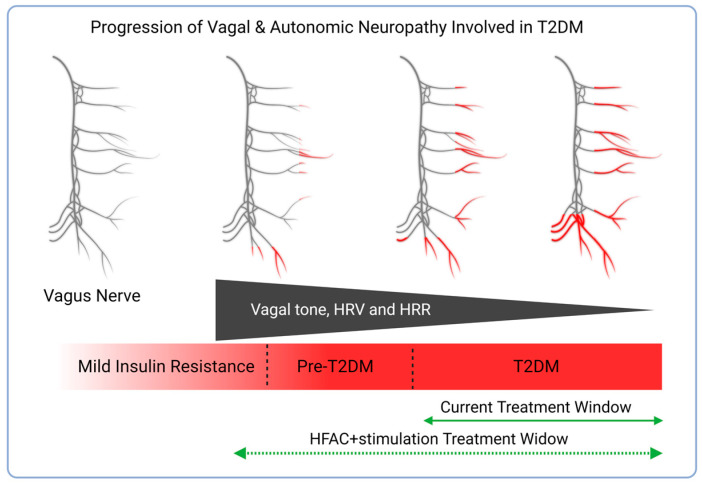
HFAC + stimulation may have a large therapy window. Diabetic neuropathy initiates at the distal ends of nerves and moves centrally during T2DM progression (neuropathy progression shown in red on the depiction of the vagus nerve). Coinciding with vagus nerve degeneration is decreased vagal tone, HRV and heart rate recovery (HRR). Current pharmaceutical therapies typically start at the onset of T2DM (i.e., FPG ≥ 126 mg/dL). Created with biorender.com (accessed on 9 August 2023).

**Figure 7 biomedicines-11-02452-f007:**
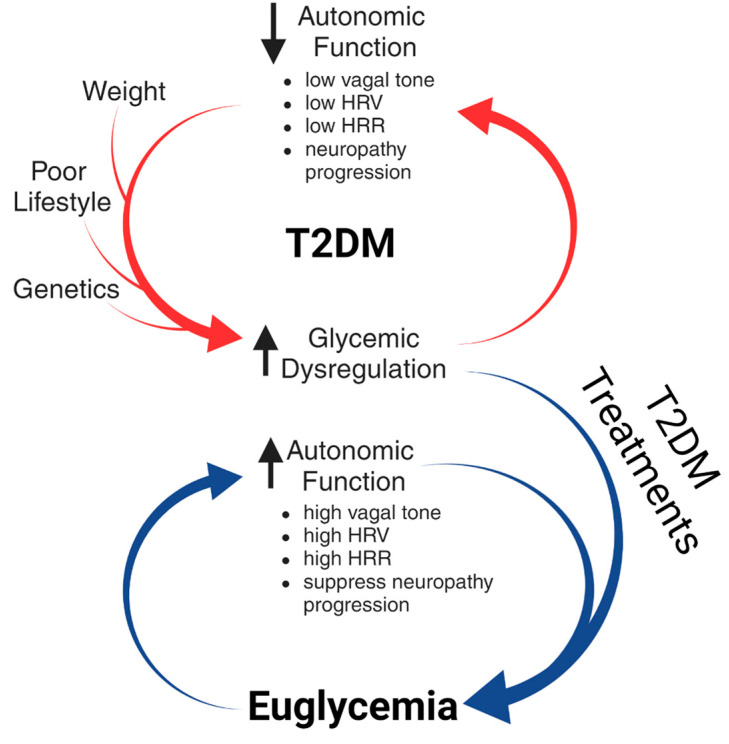
Autonomic neuropathy proceeds development of T2DM and progression of T2DM further promotes autonomic neuropathy. In parallel with autonomic dysregulation, other factors such as excess weight, little exercise, poor diet and genetics contribute to T2DM development. Methods to treat glycemic dysregulation associated with T2DM reverse autonomic dysfunction which promotes euglycemia. HRF: heart rate frequency, HRR: heart rate recovery Created with biorender.com (accessed on 29 July 2023).

**Table 1 biomedicines-11-02452-t001:** GLP-1 RA Non-Adherence Rates. Multiple studies have demonstrated that there are low adherence rates using GLP-1 RA medication even with once weekly injections. This is a common occurrence with T2DM treatments with as many as 50% of type 2 diabetics not taking medication as prescribed which is predominately due to forgetfulness [33].

Study	GLP-1 Receptor Agonist	Adherence
Divino et al. 2019 [34]	Dulaglutide	36.8–67.2%
	Exenatide twice daily	5.9–44.4%
	Exenatide once weekly	24.7–44.2%
	Liraglutide	22.2–57.5%
	Lixisenatide	15.5–40.0%
Johnston et al. 2014 [35]	Exenatide once weekly	78.3%
	Exenatide twice daily	50%
	Liraglutide once daily	72.2%
Uzoigwe et al. 2021 [36]	Semaglutide once weekly	67.0%
	Dulaglutide	56.0%
	Liraglutide	40.4%
	Exenatide one weekly	35.5%
Weiss et al. 2020 [28]	Once daily GLP-1RA	43.8%
	Once Weekly GLP-1 RA	64.2%
Weiss et al. 2022 [29]	Once daily GLP-1RA	59.8%
	Once Weekly GLP-1 RA	82.1%
	Once daily GLP-1RA	55.3%
	Once Weekly GLP-1 RA	74.1%

## Data Availability

Data not available due to proprietary reasons.

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
