# Peer review of "Combining Celiac and Hepatic Vagus Nerve Neuromodulation Reverses Glucose Intolerance and Improves Glycemic Control in Pre- and Overt-Type 2 Diabetes Mellitus"

_biomedicines, 2023, doi:10.3390/biomedicines11092452_

Round 1

Reviewer 1 Report

The manuscript by Jonathan J Waataja et al. investigated the therapeutic effect of combining celiac and hepatic vagus nerve neuromodulation on glucose intolerance in T2D. The study is of interest to the readers. I have the following suggestions:

1, the authors should revise the abstract to make it more concise. "For  example, autonomic neuropathy......." This sentence should be deleted. It is not suitable in the abstract. 

2, Table 1 should be in a three-line format. 

3, I did not see any statistical leablings in Figures 4 and 5. For example,  * p < 0.05; ** p < 0.01; *** p < 0.001. The authors must revise. 

4, Were there any changes of the body weight after implantation of the device? This should be discussed and relevant results should be presented. 

5, The authors should also check the plasma isulin levels. The prsent study is too preliminary. 

6, the authors should also discuss the limitations of the study. 

Minor editing of English language required. Some sentences are too long and complex. 

Author Response

We appreciate the reviewer’s constructive criticism, which helps us further distill the scientific merit of our manuscript.

1, the authors should revise the abstract to make it more concise. "For  example, autonomic neuropathy......." This sentence should be deleted. It is not suitable in the abstract.

We agree that the abstract should be more concise, and we have revised.

2, Table 1 should be in a three-line format. 

The table has been revised.

3, I did not see any statistical leablings in Figures 4 and 5. For example,  * p < 0.05; ** p < 0.01; *** p < 0.001. The authors must revise.

The graphs have been revised with statistical labeling.

4, Were there any changes of the body weight after implantation of the device? This should be discussed and relevant results should be presented.

There was no change in weight and this will be added to the results.

5, The authors should also check the plasma insulin levels. The present study is too preliminary.

Future studies will measure hormones such as insulin, glucagon, c-peptide and GLP-1 during IVGTTs and OGTTs.

6, the authors should also discuss the limitations of the study.

We agree there are limitations to this study (such as insulin measurements) and we highlight these in the discussion which reads “There are limitations to this current study. First, the number of animals studied was relatively small (n=6). Future research will include highly powered longitudinal studies for safety and efficacy durability. Also, this study will serve as a basis ensuring appropriate statistical power for future research. Second, this study did not explore mechanisms behind the effects of HFAC+stimulation. Future studies will measure hormones such as insulin, glucagon, c-peptide and GLP-1 during IVGTTs and OGTTs. Finally, HRV and HRR were not measured. These cardiac events will be assessed in future work.” 

Reviewer 2 Report

There are various key comments to be considered by the authors, a major revision is thus recommended.

Comment 1. Abstract:
(a) Refer to the journal’s template, and limit the wording to 200 words.
(b) Elaborate on the research results and implications.
Comment 2. Enhance the keywords “vagus nerve”, “vagus nerve stimulation”, and “heart rate variability and vagal tone”.
Comment 3. Ensure proper spacing. For examples: T2DM[1-4] and dysfunction[9-11].
Comment 4. Section 1 Introduction:
(a) Enhance the literature review with a clear summary of the methodology, results, and limitations of the latest studies (mainly recent five years journal articles).
(b) Clearly state the research contributions.
(c) Table 1 is formatted as a figure, which is inappropriate. Please refer to the journal’s template.
Comment 5. Section 2. Materials and Methods:
(a) Avoid lengthy wordings in the captions of the figures.
(b) What are the arrows in Figure 2?
(c) More details should be given for the analysis.
Comment 6. Section 3. Results:
(a) Add an introductory paragraph before Subsection 3.1.
(b) Ensure good resolutions of all figures.
(c) Figure 4, captions should include descriptions related to subfigures (a) and (b).
(d) Limited samples are taken in the results.
Comment 7. Section 4. Discussion:
(a) Comparison should be made with existing works.
(b) Elaborate on the concept in Figure 6.
Comment 8. Discuss future research directions.

Minor spell check should be performed.

Author Response

We appreciate the reviewer’s constructive criticism, which helps us further distill the scientific merit of our manuscript.

Comment 1. Abstract:
(a) Refer to the journal’s template, and limit the wording to 200 words.

We have revised the abstract and the word count is under 200 words.

(b) Elaborate on the research results and implications.
We have addressed this in our revised abstract

Comment 2. Enhance the keywords “vagus nerve”, “vagus nerve stimulation”, and “heart rate variability and vagal tone”.

We feel the words are clear and have added the following key words…..

Comment 3. Ensure proper spacing. For examples: T2DM[1-4] and dysfunction[9-11].

Proper spacing will be updated in next submission.

Comment 4. Section 1 Introduction:
(a) Enhance the literature review with a clear summary of the methodology, results, and limitations of the latest studies (mainly recent five years journal articles).

This is a great point and we have expanded the introduction to included latest research in the field.
(b) Clearly state the research contributions.

(c) Table 1 is formatted as a figure, which is inappropriate. Please refer to the journal’s template.
we have reformatted the table.

Comment 5. Section 2. Materials and Methods:
(a) Avoid lengthy wordings in the captions of the figures.

We have decreased caption sizes.

Will decrease caption size in revised manuscript
(b) What are the arrows in Figure 2?

We have revised the figure with a legend explaining the arrows.
(c) More details should be given for the analysis
.

We have added more details to analysis

Comment 6. Section 3. Results:
(a) Add an introductory paragraph before Subsection 3.1.

We have added an introduction paragraph to section 3.1
(b) Ensure good resolutions of all figures.

The resolution to the figures has been increased.

(c) Figure 4, captions should include descriptions related to subfigures (a) and (b).

Will be addressed in revised manuscript
(d) Limited samples are taken in the results.
In the future we will include highly powered longitudinal study for safety and efficacy durability.  This current study will serve as a basis ensuring appropriate statistical power for these future experiments.

Comment 7. Section 4. Discussion:
(a) Comparison should be made with existing works.’

We have added how this study compares to existing work.

(b) Elaborate on the concept in Figure 6.
We have increased elaboration for figure 6.

Comment 8. Discuss future research directions.

Future directions have been added to the manuscript predominantly in discussing the studies’ limitations

Round 2

Reviewer 1 Report

The authors have revised the manuscript accordingly. It can be considered for publication. 

Author Response

Thank you for your review of our manuscript.  We hope this adds insights to the field of neuromodulation as a treatment for type 2 diabetes, 

Reviewer 2 Report

It can be seen from the revised article that the authors have significantly enhanced the quality of the article. There are some minor comments to be followed up by the authors.

Comment 1. The main text does not follow the format in the journal’s template.
Comment 2. Ensure good resolutions of all figures. Enlarge the figures to 200%/300% to confirm that no content is blurred.
Comment 3. What is the improvement of the proposed work compared with existing works?
Comment 4. Elaborate on future research directions.

Author Response

Comment 1. The main text does not follow the format in the journal’s template.

The revised text is in the journal’s template.

Comment 2. Ensure good resolutions of all figures. Enlarge the figures to 200%/300% to confirm that no content is blurred.

We find no problems in resolution of any figure at 300%

Comment 3. What is the improvement of the proposed work compared with existing works?

Improvements are found in the paragraph starting on line 299 in the revised manuscript

Comment 4. Elaborate on future research directions.

We have highlighted future directions that are found in the limitations paragraph starting on line 305 as well as at the end of the conclusions section in the revised manuscript.